

# The impact of model resolution on the simulated Holocene retreat of the Southwestern Greenland Ice Sheet using the Ice Sheet System Model (ISSM)

Joshua K. Cuzzone[1,2], Nicole-Jeanne Schlegel[2], Mathieu Morlighem[1], Eric Larour[2], Jason P. Briner[3], Helene Serousi[2], Lambert Caron[2]

[1]University of California, Irvine, Department of Earth System Science, Croul Hall, Irvine, CA 92697-3100, USA

[2]Jet Propulsion Laboratory, California Institute of Technology, 4800 Oak Grove Drive MS 300-323, Pasadena, CA 91109-8099, USA

[3]Department of Geology, University of Buffalo, Buffalo, NY USA

*Correspondence to*: Joshua K. Cuzzone (Joshua.K.Cuzzone@jpl.nasa.gov)

**Abstract**

Geologic archives constraining the variability of the Greenland Ice Sheet (GrIS) during the Holocene provide targets for ice sheet models to test sensitivities to variations in past climate and model formulation. Even as data-model comparisons are becoming more common, many models simulating the behaviour of the GrIS during the past rely on meshes with coarse horizontal resolution (≥10 km). In this study, we explore the

impact of model resolution on the simulated nature of retreat across Southwestern Greenland during the Holocene. Four simulations are performed using the Ice Sheet System Model (ISSM); three which use a uniform mesh and horizontal mesh resolutions of 20 km, 10 km, and 5 km and one non-uniform mesh with resolution ranging from 2 km to 15 km. We find that the simulated retreat can vary significantly between models with different horizontal resolutions based on how well the bed topography is resolved. In areas of

low topographic relief, horizontal resolution plays a negligible role in simulated differences in retreat, with each model instead responding similarly to surface mass balance (SMB) driven retreat. Conversely, in areas where the bed topography is complex and high in relief, such as fjords, the lower resolution models(10 km and 20 km) simulate unrealistic retreat driven as ice-surface lowering intersects bumps in the bed topography which would otherwise be resolved as troughs using the higher resolution grids. Our results highlight the

important role that high resolution grids play in simulating retreat in areas of complex bed topography, but also suggest that models using non-uniform grids can save computational resources through coarsening the mesh in areas of non-complex bed topography where the SMB drives retreat. Additionally, these results



emphasize that care must be taken with ice sheet models when tuning model parameters to match reconstructed margins, particularly for lower resolution models in regions where complex bed topography is poorly resolved.

## 5   1 Introduction

As the cryosphere community continues to make strides in understanding processes that govern variability of the present-day ice sheets, geologic proxies constraining past ice sheet change provide important clues as to how ice sheets may have responded to past climate change (Alley et al., 2010). Decades of research have led to the development of high-resolution geologic reconstructions that detail the spatial pattern and rate of

retreat of the Greenland Ice Sheet (GrIS) over the last deglaciation as it evolved towards its present-day geometry (Weidick, 1968; Bennike and Bjorck, 2002; Young and Briner, 2015).

Southwestern Greenland is an area that experienced a large reduction in ice sheet extent. The ice margin retreated on the order of 150 km inland from the present-day coastline in response to warming during the

early and middle Holocene (Briner et al., 2016). This landscape is punctuated by widely traceable moraine sequences (Weidick, 1968; Ten Brink and Weidick, 1974) that extend nearly 600 km throughout western Greenland and provide a constraint on the past retreat pattern of the GrIS in this region; the chronology of these moraines continues to be refined (Weidick, 2012; Young et al., 2013; Larsen et al., 2015; Lesnek and Briner, 2018). This history provides a benchmark for ice sheet model-data comparisons that will further

enhance our understanding of the processes that influenced GrIS variability during the past, while at the same time will help to highlight deficiencies in existing model frameworks.

Currently, ice sheet models simulating the evolution of the GrIS are focused either on long term spinups over a glacial cycle (Huybrechts, 2002; Applegate, 2012), or its evolution during the last deglaciation (Tarasov

and Peltier, 2002; Simpson et al., 2009; Lecavalier et al., 2014; Buizert et al., 2018, Lecavalier et al., 2017). While many of these studies were primarily concerned with capturing the overall mass changes of the GrIS, one lineage of studies incorporated datasets of past GrIS change to develop a data constrained model of its evolution over the last deglaciation. This was achieved by pairing an ice sheet model with a glacial isostatic adjustment and relative sea-level model (Tarasov and Peltier, 2002; Simpson et al., 2009; Lecavalier et al.,

2014; Lecavalier et al., 2017). By incorporating data constraining the location of the GrIS beyond the present-day coastline, its vertical extent through time (i.e. ice thinning records), and records of relative sea-level, the studies by Lecavalier et al. (2014, 2017) represent the most comprehensive model of GrIS change



during the last deglaciation, with the results recently being compared against geologic archives of ice margin change (Larsen et al., 2015; Young and Briner, 2015; Sinclair et al, 2016).

While the models of Lecavalier et al. (2014, 2017) capture well the timing and retreat pattern associated with
the deglaciation in many locations, large mismatches occur, particularly in southwestern Greenland and in areas where fast flow may have dominated (Young and Briner, 2015; Sinclair et al., 2016). The climatology used to force ice sheet models through time remains a primary source for uncertainty, and great strides have been made to improve our understanding of the past climate history in Greenland through improved reconstructions of temperature (e.g., Kobashi et al., 2017; Lecavalier et al., 2017) and methods involving
data assimilation of paleoclimate proxies with climate model output (Hakim et al., 2016; Buizert et al., 2018). An area within paleoclimate ice sheet modeling that remains not well tested, however, is the sensitivity of simulated ice retreat to the ice flow dynamics model (i.e. the level of complexity in its numerical approximations) and to model resolution, both in time and space.

With regards to model setup, the use of coarse model resolutions (≥10 km grid spacing) might explain some of the model-data discrepancy (Larsen et al., 2015; Young and Briner, 2015; Sinclair et al., 2016). Driven by how well models resolve subglacial topography, simulations for the present day GrIS have shown an important dependence of model resolution on accurately simulating ice flow across Greenland (Greve and Herzfeld, 2013; Aschwanden et al., 2016). Dependence of model resolution also extends to modeling future
ice mass loss, where higher-resolution models simulate more mass loss than models with lower resolutions (Greve and Herzfeld, 2013). Although some work has focused on model resolution and its impact on simulated mass flux from the GrIS for the present day and future, how model resolution affects simulated retreat in paleo-ice-sheet-modeling studies is not well constrained. Prior work demonstrates that low resolution grids limit a model's ability to capture features such as ice streams and marine terminating outlet
glaciers (Aschwanden et al., 2016), which might be on the order of a few kilometers in width. Additionally, in land terminating portions of an ice sheet, low model resolution may lead to large jumps in the snowline, which ultimately can lead to large advances or retreat in the ice margin on the order of the model resolution (Young and Briner, 2015; Sinclair et al., 2016), and therefore limit the ability to capture smaller-scale ice marginal fluctuations (i.e. km scale).


In this study, we present results from regional ice sheet modeling experiments in Southwestern Greenland over the Holocene using the three-dimensional thermomechanical Ice Sheet System Model (ISSM). We build on earlier efforts that focused on this ice sheet sector (e.g., Van Tatenhove et al., 1995). Ice model resolution

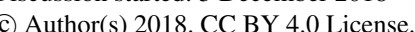



is the primary target for assessment here, with four separate simulations being run, each with its own horizontal resolution ranging from 20 km to 2 km. Since model resolution is a constraint that is typically not explored when studying the past due to the computational cost, in this study, we aim to determine whether increased model resolution is worth the computation time for simulating past ice sheet retreat.

## 2 Model description and setup

### 2.1 Ice sheet model

We use the Ice Sheet System Model (ISSM; Larour et al., 2012), a finite element, thermomechanical ice sheet model. We choose the higher-order approximation of Blatter (1995) and Pattyn (2003), herein referred to as

BP, to solve the momentum balance equations. Although this ice flow approximation is typically not used to model ice sheets over paleoclimate timescales, our choice is based on representing the past dynamics of the ice sheet history as best as possible even though computational time is increased over conventional paleoclimate ice sheet models using the more common shallow ice approximation (SIA; Hutter, 1983).

The model domain for this study (**Figure 1**) focuses on the southwestern region of Greenland where geologic proxies detail Holocene ice retreat from the present-day coastline (Weidick, 1968; Ten Brink and Weidick, 1974). By constraining our domain to southwestern Greenland, the number of mesh elements within the model can be minimized when compared to modeling the entire GrIS, therefore reducing computational load. The model domain extends from the present-day coastline to the ice sheet divide. The southern and northern

borders of the domain coincide with areas of minimal north-to-south across-boundary flow based upon present-day ice-surface velocities from Rignot and Mouginot (2012). The associated boundary conditions used to drive the model are discussed in section **2.6.**

### 2.2 Domain discretization

Typically, prior paleoclimate ice sheet modeling efforts across Greenland have used uniform meshes with a horizontal resolution of 20 km (Simpson 2009; Lecavalier, 2014); a more recent model used a 10 km horizontal mesh resolution (Buizert et al., 2018). Because many of the geologic archives constraining past margin behavior reside in areas of complex topography, coarse-resolution models do a poor job of capturing the complexities of the underlying topography, which is one condition important for capturing contemporary

ice flow (Aschwanden et al., 2016).

The first three models are generated using a uniform triangular grid, with horizontal resolutions of 20 km, 10 km, and 5 km. The fourth model (herein referred to as *non-uniform*) relies on anisotropic mesh adaptation,



whereby the element size varies as a function of the bed topography. The maximum horizontal mesh resolution is 15 km in areas of smooth bed topography (primarily over the interior of the domain) and becomes progressively finer in areas of high relief, with a minimum horizontal resolution of 2 km (mainly in fjord regions). The bed topography for each model is taken from BedMachine Greenland v3 (Morlighem et

al, 2017) and is initialized with present-day ice surface elevation from the GMIP DEM of Howat et al. (2014). In **Figure 2**, the corresponding bed height is shown for each model detailing the associated differences based on horizontal grid resolution. Generally, the bed topography is captured better using the higher resolution mesh, with the non-uniform mesh (**Figure 2A**) being able to best resolve valleys along the present-day coastline. The 5 km mesh captures the same general topographic features as the non-uniform mesh albeit

with less detail. At 10 km, individual valleys become unresolved, particularly around Nuuk and Jakobshavn Isbræ (see **Figure 1** for locations). The 20 km model fails to capture any topographic features that would hold glacier outlets.

### 2.3 SMB (SMB)

We use the positive degree day method outlined in Tarasov and Peltier (1999) to construct the necessary accumulation and ablation history used to drive our ice sheet model during the past. The spatial monthly mean surface air temperature and precipitation climatology spanning the period 1980-2010 is taken from Box et al. (2013). The surface air temperatures are then scaled based upon isotopic variations in the Greenland Ice Core Project (GRIP) $\delta^{18}O$ record (Dansgaard et al., 1993) as follows:

$$\Delta T(t) = d(\delta^{18}O(t) + 34.83) \tag{1}$$

where d = 2.4°C‰$^{-1}$ (Huybrechts, 2002). Anomalies from equation 1 are applied to the present-day climatology to create a temperature forcing back through time. Deviations in the precipitation climatology

arise from a purely thermodynamic relationship as precipitation rate changes 7.3% for every 1°C of temperature change derived in equation 1. For the positive degree day scheme, snow melts first (0.006 m °C$^{-1}$ day$^{-1}$) followed by bare ice (0.0083 m °C$^{-1}$ day$^{-1}$), with allocation for superimposed ice included. The temperature forcing is adjusted throughout the run using a lapse rate correction of 5°C km$^{-1}$ (Abe-Ouchi et al., 2007) to account for changes in ice surface height throughout the simulation, while elevation-dependent

desertification is included (Budd and Smith, 1981) to ensure reduction in precipitation by a factor of 2 for every kilometer change in ice sheet surface elevation. Further details regarding the positive degree day scheme implemented within ISSM can be found in Le Morzadec et al. (2015).





### 2.4 Thermal model and basal drag

The thermal evolution of the ice is captured using an enthalpy formulation described in Aschwanden et al., (2012), which includes formulations for both temperate and cold ice. Surface air temperatures are imposed at the ice surface transiently, while geothermal heat flux (from Shapiro and Ritzwoller, 2004) is applied at

the base. The model contains 5 vertical layers, with spacing between layers decreasing modestly towards the base. To simulate the vertical distribution of temperature within the ice sheet, we rely on quadratic finite elements (i.e. P1xP2) along the z-axis as a means for our vertical interpolation. Details of the implementation and description of these higher-order vertical finite elements can be found in Cuzzone et al. (2018). Through using higher-order finite elements as a means for vertical interpolation, this method allows the ice sheet

model to capture sharp thermal vertical gradients particularly at the bed, which is an improvement over conventional methods using a linear vertical interpolation, despite having fewer vertical layers. This ultimately limits the necessity for a large number of vertical layers in our ice model and therefore decreases computational load.

To capture spatial variations in sliding, the spatially varying basal drag coefficient ($k$) in equation 2 is derived using inverse methods (Morlighem et al., 2010; Larour et al., 2012), providing the best match between modeled and InSAR surface velocities (Rignot and Mouginot, 2012).

$$\tau_b = -k^2 N^r ||V_b||^{s-1} v_b \ , \tag{2}$$

where the $\tau_b$ represents the basal stress, $N$ represents effective pressure, $V_b$ represents magnitude of the basal velocity, and $r$ and $s$ are friction law exponents, set equal to 1 for these experiments.

Since the drag coefficient ($k$) derived using this methodology is constrained to modern day, we adopt an

approach based upon Hindmarsh and LeMeur (2001) and Greve (2005) to construct a spatially varying temperature dependent scaling parameter ($\lambda$) as a function of time.

$$\lambda_t = e^{(T_{b(modern)} - T_{b(t)})/\alpha} \ , \tag{3}$$

where, $T_{b(modern)}$ is the basal temperature relative to pressure melting derived from a thermal steady-state computation for modern day (Seroussi et al., 2014), $T_{b(t)}$ is the basal temperature relative to pressure melting at time $t$, and $\alpha$ is a constant scaling factor (°C) often referred to as the sub-melt parameter (Hindmarsh and Le Meur, 2001). For these simulations $\alpha$ is set equal to 5. This number was chosen as it allows for a Last





Glacial Maximum (LGM) GrIS simulated ice volume that is consistent with other ice sheet models that restrict ice extent only to present day land (Applegate et al., 2012; Robinson et al., 2011). It is noted that values for this parameter lack a theoretical basis (Hindmarsh and Le Meur, 2001), and are often set to a value that prevents numerical instabilities from arising. Lastly, we scale the spatially varying basal drag coefficient

($k$), as a function of $\lambda$ with maximum values capped at 300 to limit numerical instabilities that may arise from unreasonably large numbers:

$$\tau_b = - \min(300, \lambda_t)\, k^2 N^r ||V_b||^{s-1} v_b \qquad (4)$$

For this approach, the basal stress ($\tau_b$) increases as the basal temperatures decrease relative to present day, with virtually no sliding occurring for high values of $k$. Conversely, the basal stress $\tau_b$ decreases as basal temperatures increase, with high sliding for low values of $k$. Lastly, the ice hardness, B, is temperature dependent following the rate factors given in Cuffey and Paterson (2010, p. 75). We initialize B by solving for a present-day thermal steady state (Cuzzone et al., 2018), while during forward runs, B evolves transiently

through time.

### 2.5 Experimental setup and boundary conditions for regional domain

At the ice front, we impose a free-flux boundary condition, and Dirichlet boundary conditions for the southern, northern and ice divide boundaries. To create the necessary transient boundary conditions (ice

thickness, temperature, and velocity), we perform a continental-scale GrIS simulation from the last glacial maximum (21,500 years ago) to present day. This continental-scale simulation uses the BP ice flow approximation and is performed on a 5-layer non-uniform mesh ranging in horizontal resolution from 3 km in areas of high present-day surface velocities to 20 km over the ice interior. It is performed using forcings and parameterizations similar to the regional model, as described in sections **2.3**, **2.4**, and **2.5**.

For the regional model, we initialize the model with present day geometry and run a relaxation centered at 12,000 years ago, applying the appropriate interior ice boundary conditions of ice thickness, ice temperature, and the x and y component of ice velocity from the continental-scale GrIS simulation. This time period is chosen as the ice margin over southwestern Greenland was near or at the present-day coastline (Young and

Briner, 2015). The relaxation runs 20,000 years until the ice volume is in equilibrium. From here, the models are run transiently to present day. Since ISSM currently does not have capabilities to model solid earth viscoelastic deformation transiently, we include an offline time-dependent forcing that accounts for changes in relative sea level from glacial isostatic adjustment (Caron et al., 2018), which modifies the land area



available for glaciation and impacts the presence of floating ice. While grounding line migration is simulated in these experiments, calving and submarine melting of floating ice is not included.

### 2.6 Present day thermal steady state ice surface velocities

The thermal steady state ice surface velocities for present day are shown for each individual model (**Figure 3**). Generally, representation of faster ice flow along the coast improves with increasing resolution (i.e. increasing RMSE for lower resolution models compared to Rignot and Mouginot (2012)). This is primarily attributed to an improved representation of subglacial topography and ice thickness in the more highly resolved models (Aschwanden et al., 2016). As many of the outlet glaciers along this margin have troughs that are on the order of a few kilometers in width, the lower resolution models (10 km and 20 km) do not fully resolve the fast-flowing ice streams of Jakobshavn Isbrae and outlets to its north. Outlet glaciers in the southern portion of the domain near Kangiata Nunâta Sermia (KNS) are also less well resolved in the lower resolution models, although the general swath of higher velocities is captured well for most fast-flowing areas of the ice sheet when compared to the observations (Rignot and Mouginot, 2012). It is noted that the non-uniform mesh represents these faster flow features best when compared to observations in most regions due to its high resolution. Accordingly, the associated mass flux at the ice margin is representative of these differences in model resolution, with the 10 km and 20 km models having approximately a 25% increase in mass flux (GT/yr) compared to the observations, while the 5 km and non-uniform mesh have approximately 5 to 9% increase in mass flux compared to observations.

## 3 Results

### 3.1 Relaxed state at 12 ka

The models are relaxed for 20,000 years using a constant climate corresponding to 12 ka (**Figure 4**). The four models simulate increasing ice volume with decreasing model resolution; the 20 km model simulates approximately 6% less total ice volume than the non-uniform model. Ice surface velocities for the relaxed states (**Figure 5**) depict the role of model horizontal resolution in capturing fjords and narrow outlets close to the model domain edge (i.e. present-day coastline). Generally, the two higher-resolution models (non-uniform and 5 km) capture narrow, fast flow in these outlets, whereas the lower resolution models simulate a more diffuse pattern in the ice surface velocities. This is primarily the case in areas where the bed topography is better resolved in the higher-resolution models, and therefore confines the flow to narrow outlets. Ice velocities are reduced in the higher-resolution models for areas where the low bed topography



that channels ice flow is interrupted by bumps and depressions in the bed. These features become less resolved in the lower resolution models, with the 20 km model simulating much higher ice surface velocities in the Nuuk and Jakobshavn areas. Consistently, ice mass flux (in GT/yr) along the ice front (at the present-day coastline) is 34% and 14% higher than the non-uniform model for the 20 km and 10 km model, respectively, which is the primary driver for lower simulated ice volumes for the relaxed 12 ka state.

### 3.2 Simulated ice volume (12 ka to present day)

The ice volume evolution for each model is shown in **Figure 6A** along with the Min-Max normalized ice volume shown in **Figure 6B**. Generally, the 20 km and 10 km models simulate the lowest present-day ice volumes, but they also begin at 12 ka with lower ice volumes than the higher-resolution models. Each model follows a similar trend, with ice volume loss occurring between 12 ka and 1 ka, followed by an increase in ice volume to present day. During ~10 ka and 5 ka the relative rate of ice mass loss increases for the higher-resolution models (**Figure 6B**), which is likely related to better resolved ice flow, particularly in the outlets near Nuuk (see results in section **3.4**).

### 3.3 Large scale simulated retreat

**Figure 7** shows the simulated extent at 11.2 ka, 10.5 ka, and 9.5 ka for each model. All four models generally show ice retreat from the coastline occurring between 11.5 ka and 11.2 ka in the northern portion of the model domain, whereas the ice margin experiences little to no retreat farther south. Despite differences in horizontal resolution, all models show a similar magnitude and pattern of retreat, with higher-resolution models depicting details in the ice margin similar in scale and sinuosity to the mapped pattern of moraines. Similarity of the magnitude and pattern of retreat also occurs at 10.5 ka and 9.5 ka amongst all models. In contrast to the northern portion of the model domain, the southern portion features a simulated retreat that varies widely based upon model resolution. For example, at 10.5 ka, the higher-resolution models (5 km and non-uniform) exhibit little retreat from the coastline, whereas the 10 km and 20 km models show upwards of 50-60 km of retreat. Differences in retreat between the higher- and lower-resolution models are further seen at 9.5 ka. Over land-terminating portions in the southern area, the modeled ice margin retreats similarly within all models; however, in the fjord regions (e.g. inland of Nuuk), only the 10 km and 20 km models show ice margin retreat (of ~50-70 km) whereas the higher-resolution models exhibit no ice margin retreat.

### 3.4 Simulated retreat (12 ka to present day) – along flowline

To better illustrate simulated ice margin behavior through time, we analyze ice retreat along five specific flow lines (A through E) across the domain (**Figure 8**). In **Figure 8**, ice retreat along flowline **A** is shown



for each model. All models show a similar trend with the highest retreat rate (upwards of 50 -100 m/yr) occurring between approximately 11.5 ka and 10 ka. Between ~10 ka and the present, all simulated ice margins generally reside within 10 km of the present-day ice margin. The retreat history simulated by the 10 km and 20 km models exhibits a relatively stable ice margin for much of this period, whereas the higher-

resolution models (i.e. non-uniform and 5 km) depict an ice margin that characterized by higher-frequency variability on the order of 5-8 km. The retreat history along flowlines **B** and **C** is consistent in timing and pattern to flowline A (shown in **Figure S2** and **S3**).

Differences in the bed topography between the four models reflect model resolution, with higher-resolution

models capturing topography closer to reality (**Figure 8**). Nevertheless, the bed topography along the flowline **A** is similar among the different models owing to the low relief topography in this region. The low-angle ice surface responds to surface melt similarly among the four models along flowline A (See supplemental **Figure S1** for surface temperature and SMB along flowlines) and is likely why the retreat history is similar. Along flowline **B** (**Figure S2**), bed topography in all models exhibits increasing elevation

into the ice sheet interior. Whereas the non-uniform and 5 km models capture a trough between 30 km to 120 km along the flowline, this feature is subtle in the 10 km model and nonexistent in the 20 km model. Similar to flowline **A**, however, the simulated retreat in flowline B is similar amongst the different models both in rate and magnitude of retreat. Generally, the ice margins exhibit retreat forced primarily through surface lowering in response to negative SMB (**Figure S1**) because the ice margin retreats similarly through areas of

varying bed topography. At approximately 150 km along the flowline, the ice retreats into an area with higher bed topography, and the ice surface profiles become steeper, thereby raising the ELA and stabilizing the ice margin. Along flowline **C** (**Figure S3**), the bed topography for the non-uniform and 5 km model is characterized by low elevations at the beginning and end of the flowline, with a peak in elevation in the middle, while the 10 km and 20 km models exhibit a more consistent bed elevation along the flowline. The

upward slope of the bed for the non-uniform and 5 km model tends to slow the retreat of the ice margin along flowline C during the early Holocene, although differences between the 10 km and 20 km models are not dramatic. After 8 ka, ice retreat stabilizes in all models, similar to flowlines **A** and **B**, although the non-uniform and 5 km models exhibit ice marginal fluctuations on the order of 5-8 km.

In the southern portion of the domain, the fjords within the Nuuk region dominate the landscape. The ice margin in flowline **D** (**Figure 9**) remains fixed for the entire simulation in the non-uniform and 5 km model simulations. In contrast, both the 10 km and 20 km models depict retreat at ~9.5 ka, after which the 10 km model quickly stabilizes and the 20 km model exhibits variability up to present day. All models fail to



simulate a present-day ice margin that comes close to today's observed ice margin (**Figure 9**). The lower-resolution models simulate retreat in this region on the order of 30 to 50 km, which is controlled primarily by the bed topography. In reality, a trough extends much of the distance along this flowline, which is captured well by both the non-uniform and 5 km mesh, where depths reach ~500 m below sea level. Consequently, the non-uniform and 5 km models are better able to capture the stress balance and mass transport as they simulate more realistic ice flow and delivery of ice mass to the margin in this region. In the 10 km model, surface lowering intersects a bed bump that is above sea level at approximately 40 km along flowline **D**. Inland of this, the 10 km model bed contains a shallow trough, which is capable of sustaining the ice margin throughout the remainder of the simulation. The 20 km model lacks any clear trough and instead captures a significant rise in the bed topography at 70 km along the flowline, where the other models resolve a trough. As the ice surface lowers along flowline **D**, it becomes increasingly influenced by this bed feature. Due to the upward slope and horizontal top of this bed feature, the margin varies in response to the high-frequency climate variability during the Holocene (**Figure S1**).

Flowline **E** (**Figure S4**) follows a narrow and shallow trough south of flowline **D**. This shallow trough is only captured completely in the non-uniform model, although the 5 km model captures low topography along the ice margin. In the 10 km and 20 km models, there is no indication of a trough and instead the bed topography is high and generally flat. Similar to flowline **D**, downwasting via negative SMB (**Figure S1**) drives ice retreat in the 10 km and 20 km models. Because the non-uniform mesh captures a trough along flowline E, delivery of ice mass to the margin continues through the Holocene, stabilizing the ice margin position despite surface lowering through negative SMB.

## 4 Discussion

### 4.1 Geologic context of simulated retreat

We find that model resolution plays a negligible role in the simulated ice margin history in the northern portion of our domain along flowlines **A**, **B**, and **C**. In the southern domain along flowlines **D** and **E**, however, model resolution plays a large role in the simulated ice margin history.

In the northern portion of our model domain, geologic archives indicate retreat from the coast occurred between 12 ka and 11 ka (Kelly et al., 2015; Young and Briner, 2015), which is generally consistent with our simulations regardless of resolution. The subsequent retreat in all models towards the present-day ice margin is also generally consistent with the geologic reconstructions of ice margin retreat in this region. van Tatenhove et al. (1996) provide one of the earliest ice sheet model – data comparisons for this region (around



flowline **C**) during the last deglaciation. van Tatenhove et al. (1996) compared three different ice sheet models ranging in resolution from 20 km to 40 km to ice margin reconstructions constrained by radiocarbon ages from the region, and indicated that model resolution played little role in the inter-model retreat differences. van Tatenhove et al. (1996) pointed to the strong governing role of SMB in this region with little

influence from ice streams. Likewise, simulations from the 20-km-resolution model of Lecavalier et al. (2014) show reasonable agreement in the retreat across this region when compared to geologic reconstructions. Our results indicate that the bed topography in this region is well represented among the different models, despite their differences in resolution, and thus simulated ice margin history faithfully responds to SMB forcing and is not complicated by ice flow adjustments to underlying topography. However,

one feature that stands out in the higher-resolution models (5 km and non-uniform) is the presence of high frequency ice marginal fluctuations on the order of 5-8 km (**Figures 8**, **S2**, **S3**). The geologic record indicates that small-scale marginal fluctuations are likely responsible for the moraine record and seem to be related to high-frequency variability in temperature (e.g., Young et al., 2013). Thus, models capable of capturing small scale fluctuations in the ice margin history are valuable for comparing with geologic constraints of past ice

sheet change. The inability of lower-resolution models to capture these features has been highlighted in previous work (van Tatenhove et al., 1996; Larsen et al., 2015), and hampers data-model comparisons.

In the southern portion of the model domain, fjord systems provide a different bed setting than in the north, presenting a significant challenge for modeling ice margin change. Deep and narrow troughs up to 500 m

below sea level and 3-5 km wide seemingly played an important role in governing ice margin retreat. Many geologic archives that constrain past ice margin variability in this region (Sinclair et al., 2016), reveal rapid deglaciation from the present-day coastline to near the present-day margin at ~10 ka (e.g., Larsen et al., 2014). None of our experiments match the geologic observations.

Our simulated ice retreat is highly dependent on model resolution in this area because the different models represent the bed topography quite differently. For example, only the non-uniform and 5 km models capture the deep fjords, whereas the 10 km and 20 km models have unrealistic bed features that end up driving retreat (**Figures 9** and **S4**). Our simulations do not include calving or submarine melting, and therefore, each model's simulated ice surface responds similarly to negative SMB. However, the ability of the high-resolution models

to capture the narrow and deep fjords allows the ice margin to persist as the stress balance and mass transport is well captured. Since the fjords are not well represented in the low-resolution models, there is lower delivery of ice mass to the margin, and the simulated retreat is driven as the ice surface lowers and intersects elevated bump artifacts in the bed topography. While none of the models capture the timing or amount of retreat




accurately, the high-resolution models in this case perform the worst, capturing negligible retreat. The rapid ice margin recession recorded by the geologic reconstructions in this marine-dominated region probably highlight the influence of calving and enhanced submarine melting of floating ice, neither of which are included in our model simulations. The lack of submarine melting, in particular, may lead to the model-data
mismatch; available evidence (e.g., Dyke et al., 2014) supports the influence of the warm Irminger Current during the early Holocene, which likely penetrated fjords up to the ice margin.

There are stark differences in processes affecting retreat of the land-dominated ice margin (i.e. SMB in the northern section of the model domain) and the marine-dominated ice margin (i.e. calving and submarine melt
in the southern portion of the domain). These different drivers of ice margin change also affect different sectors of contemporary GrIS (Sole et al., 2008; Straneo and Heimbach, 2013). Our results highlight that in areas of simple, low-relief bed topography, SMB drives the simulated retreat with little differences existing between models of varying spatial resolution. Conversely, in areas with complex, high relief bed topography, such as in fjord settings, models that are unable to capture the deep and narrow troughs may unreasonably
simulate retreat (cf. Åkesson et al., 2018).

We find that a high-resolution mesh is not needed to capture margin retreat in areas where the bed topography is not complex, and efforts to match geologic reconstructions will be better served by focusing on representing SMB as accurately as possible. On the other hand, comparing simulated retreat to geologic
reconstructions in areas of complex bed topography requires model resolution capable of capturing km-scale features (i.e. narrow troughs), and highlights the need for high resolution bed maps within fjord regions. Anisotropic mesh capabilities play an important role in allowing a model to adjust its resolution spatially while using computer time efficiently.

For low-resolution models, care must be taken in these regions when attempting to capture the reconstructed retreat. For example, in order to satisfy relative sea level records used to constrain an ice sheet model of the GrIS, Lecavalier at al. (2014) artificially increased middle Holocene temperatures used to drive the ice sheet model. Although this resulted in a simulated ice margin history consistent with available geologic records, it is noted that such external forcings may drive unphysical retreat in areas of complex bed topography that
may otherwise have been driven by ice dynamics. Another consideration is the regional setting presented here. Since these experiments are focused on the southwestern GrIS, an area that may be relatively topographically uniform, we expect the results of the marine-influenced region in the southern part of our





domain to be most relevant for other portions of the GrIS, in particular eastern Greenland where fjords dominate the landscape (Morlighem et al., 2017).

### 4.2 Model limitations

5 When simulating the retreat of the southwestern GrIS, our choice of climate forcing, using the GRIP $\delta^{18}$O record (Dansgaard et al., 1993), follows what has been a cornerstone in forcing Greenland ice sheet modeling over the paleoclimate record (Huybrechts, 2002; Greve, 2011; Applegate et al, 2012). This approach has been adjusted in Tarasov and Peltier (2002), Simpson et al. (2009) and Lecavalier et al. 2014) by synthetically increasing Holocene temperatures, with more recent simulations of the deglaciation of Greenland making

10 use of more recent proxy-temperature reconstructions that are better constrained throughout Greenland (Lecavalier et al., 2017; Buizert et al., 2018). Nevertheless, using a single, scaled paleoclimate record from Summit ignores the more likely history of a spatio-temporally variable climate history spanning the Holocene around Greenland (cf. Vinther et al., 2009). In any case, since the traditional approach (i.e. the GRIP $\delta^{18}$O scaling) assumes that the spatial variability in temperature and seasonality remains fixed to modern day, our

15 results cannot fully reconcile how changes in the magnitude of warming and spatial variation of that warming affects our results.

Although our simulations have no reasonable representation of calving, the results do indicate that models with a resolution of 10 km or greater would be likely unable to address calving processes in fjords, as typical

20 fjord width is ≤10 km. Often calving in ice sheet models relates to water depth, considering past changes in eustatic and relative sea-level (Huybrechts, 2002; Simpson et al., 2009; Lecavalier et al., 2014). Making these simple parameterizations is not always possible in fjord regions where these types of data are missing. Although the high-resolution models presented here do capture the narrow fjords, implementation of a calving scheme currently would be computationally intensive. One possibility for future work would be to

25 force the model with high submarine basal melt rates as a proxy for calving, as done in Åkesson et al. (2018). Submarine melt has been shown to be an important mechanism driving both contemporary ice mass loss (Rignot, 2010) and past GrIS variability on glacial/interglacial timescales (Bradley et al., 2018; Tarbone et al., 2018). Although few constraints do exist detailing past variations in ocean temperature for the Labrador Sea (Gibb et al., 2015) and Disko Bugt (Jennings et al., 2006), applying submarine melt rates to marine

30 termini throughout our model domain would not be possible without significant uncertainty.

### 5 Conclusion



We investigated how ice sheet model resolution influences the simulated Holocene retreat of the southwestern GrIS using ISSM. Our focus on the southwestern GrIS is driven by two factors: First, the regional approach allowed for modeling at a high resolution (for the non-uniform and 5 km mesh) while minimizing computational costs that would increase significantly while running a GrIS-wide simulation.

Second, the southwestern GrIS is an area where geologic archives indicate the ice sheet underwent large-scale and relatively well-known retreat during the Holocene.

The results presented here indicate that model resolution has a selective influence in the simulated retreat over southwestern Greenland during the Holocene. In particular, simulated retreat can vary significantly

between models of different resolution based on how well the bed topography is resolved. In areas where the bed topography is relatively simple, low relief and free from marine influence, model resolution plays an insignificant role in influencing the pattern and rate of retreat. In these areas, models with different resolutions respond similarly to surface-mass-balance-driven retreat. On the other hand, in areas with complex and high relief bed topography, such as deep troughs and fjords, the low-resolution models lead to

unrealistic retreat. As all models in these simulations only respond to surface melt and therefore ice surface lowering (and no mass loss via calving or submarine melt), the low-resolution models (10 and 20 km) simulate ice retreat driven purely as a consequence of incorrectly capturing the bed geometry. As one example, ice-surface lowering in these models intersects bed bumps that would otherwise be resolved as a trough in higher-resolution models.

Our results imply that computational resources can be saved when modeling certain portions of the GrIS. Conversely, the results also highlight the importance of model resolution in areas of complex topography. Ice sheet models using a non-uniform mesh can adapt grids to fit these constraints while using computation time efficiently, but for models using uniform fine-scale meshing, resolving such features becomes

computationally difficult, especially over long paleoclimate timescales. As ice sheet models sometimes rely on the geologic record for validation, care must be taken in evaluating model-data misfits. In areas of complex topography, over tuning of model parameters or climatology may occur in low-resolution models that seek to match the reconstructed margin. We suggest that increased model resolution is critical in regions dominated by fjords (e.g. southeastern Greenland).


Future work with ISSM will focus on using a model that has lower resolution in areas driven mainly by SMB and higher resolution in areas influenced by ice dynamical processes using non-uniform mesh capabilities. Future work will also seek to evaluate the sensitivity of using improved climate forcings (Hakim et al., 2016;



Buizert et al., 2018), better representations of ice dynamics (calving and submarine melt), and more quantitative comparisons to improved ice margin reconstructions.

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

20  **Figures**

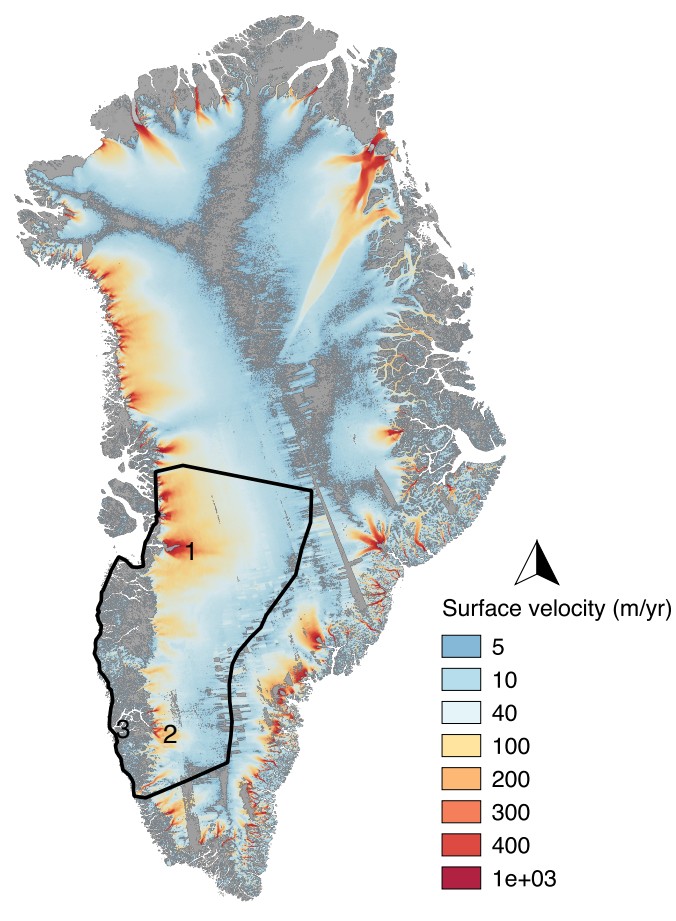

**Figure 1.** Pre... for the Greenland Ice ... ations correspond to:



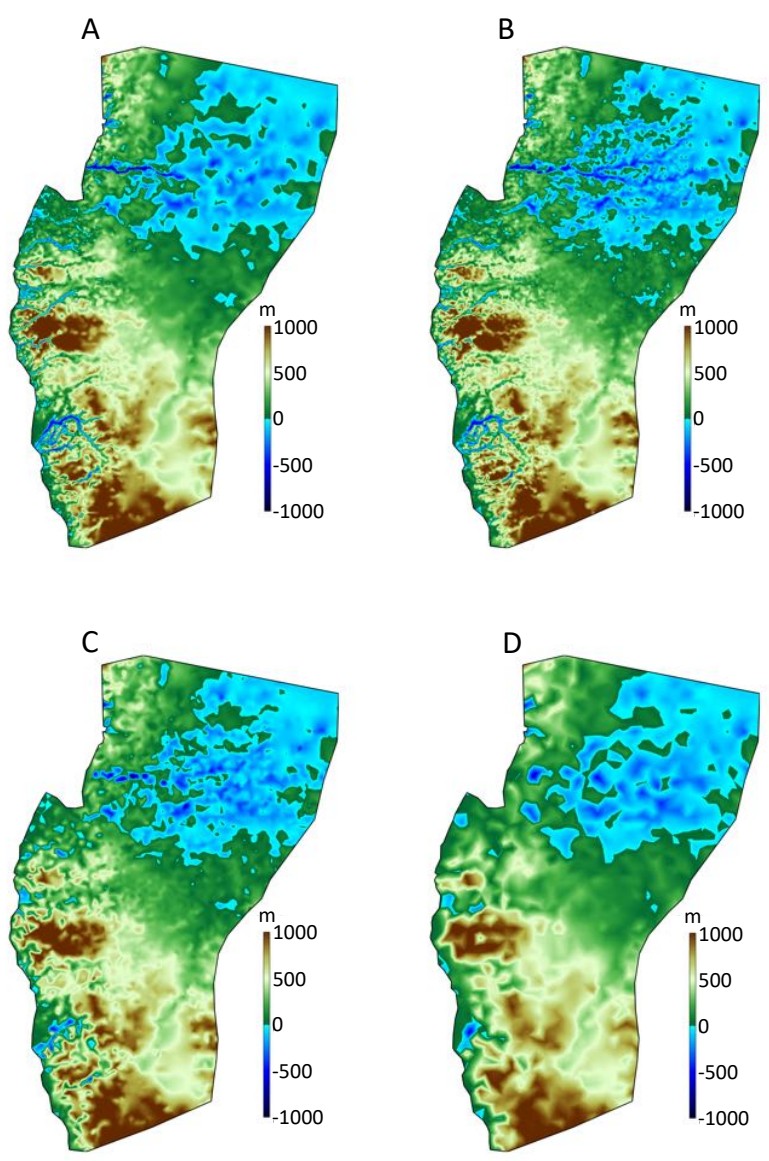

**Figure 2.** Associated bed topography maps for the non-uniform high-resolution mesh (A), uniform 5 km mesh (B), uniform 10 km mesh (C), and uniform 20 km mesh (D).





**Figure 3.** Observed ice surface velocity (Rignot and Mouginot, 2012), present day steady state ice surface velocities for each individual model, and differences from observations.





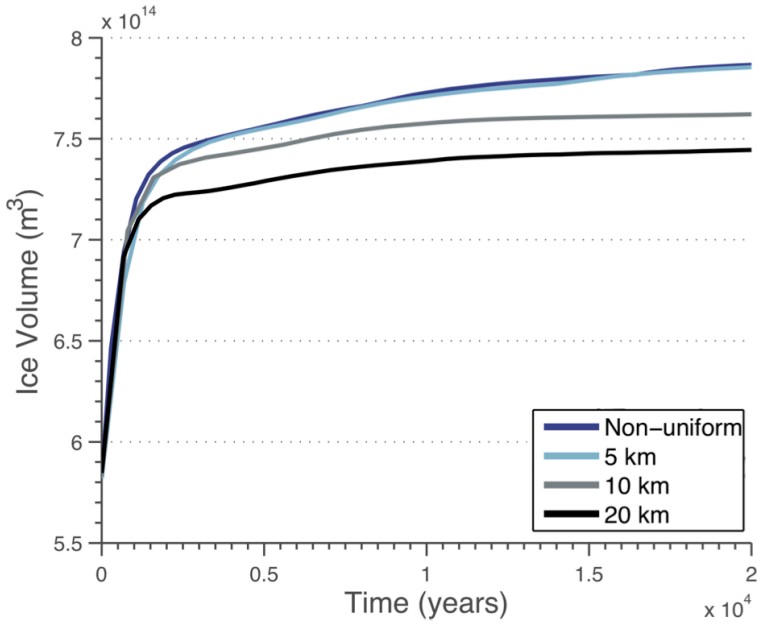

**Figure 4.** Ice volume evolution for the 12 ka constant climate relaxation.



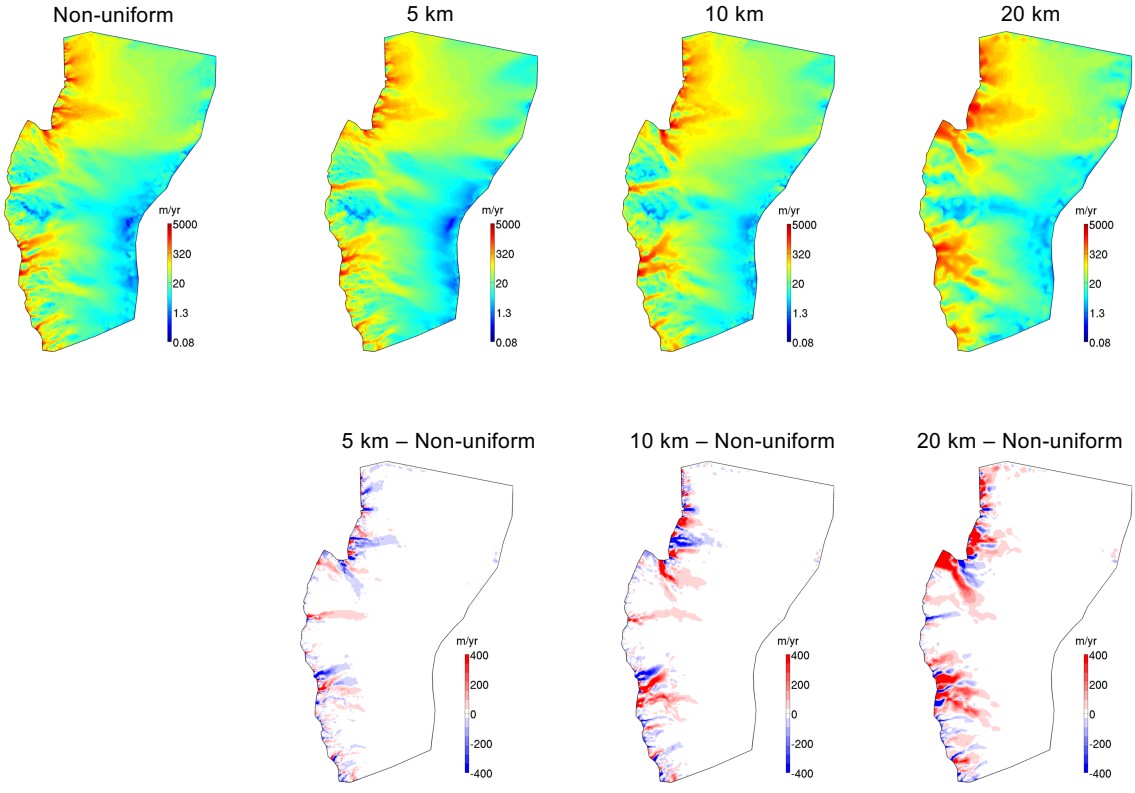

**Figure 5.** Relaxed ice surface velocities at 12 ka for each model, and differences from the non-uniform model.





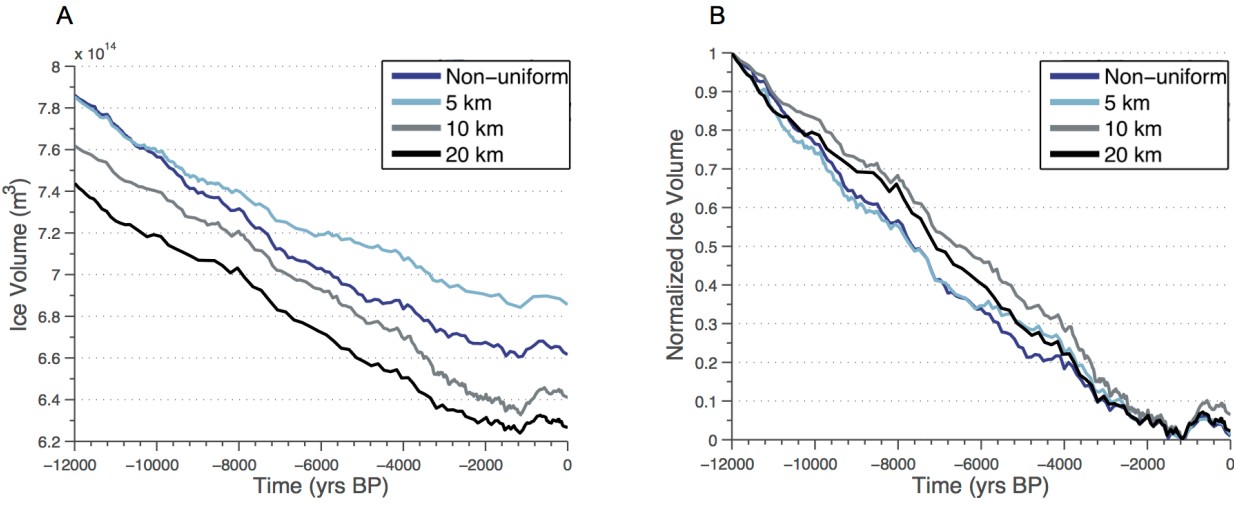

**Figure 6.** A. Transient ice volume evolution for the simulations from 12 ka to present day. B. Min-Max normalized ice volumes over the transient simulation.





**Figure 7.** Simulated ice sheet margin for the different model resolutions shown over locations in the northern (A) and southern (B) domain. The present-day ice thickness is shown, derived from Morlighem et al. (2017) and Howat et al. (2014).





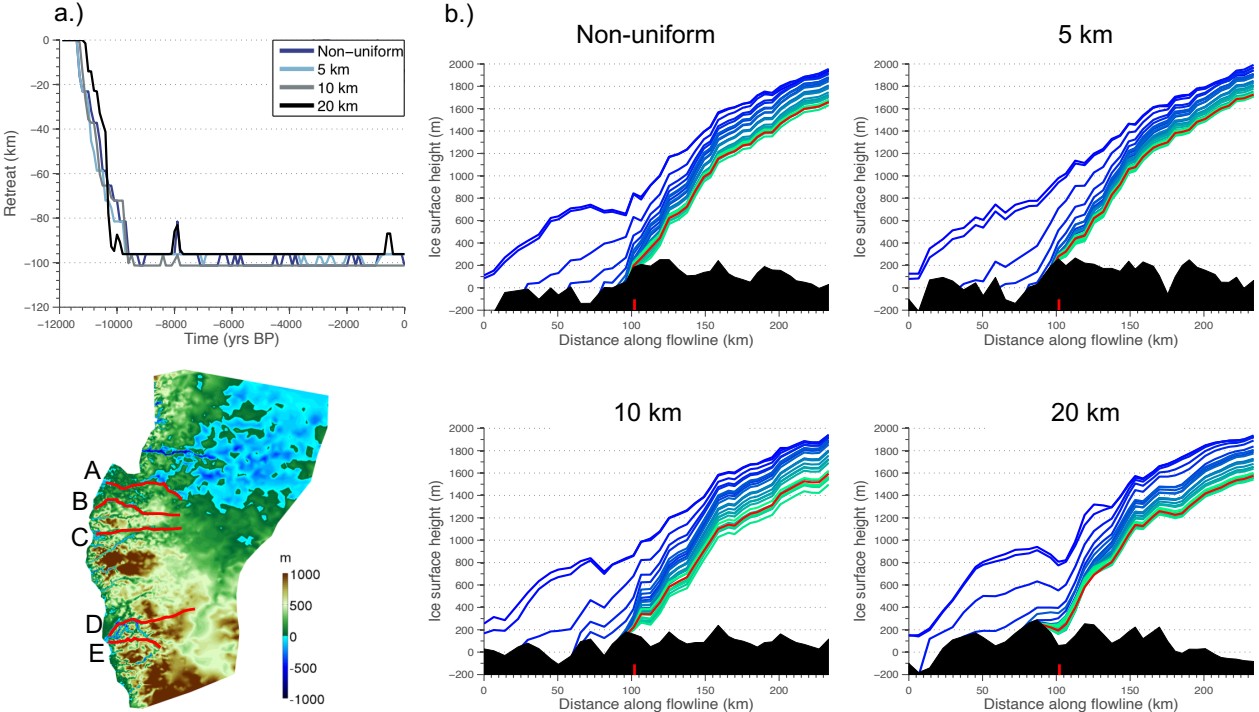

**Figure 8.** a) Simulated retreat along flowline (A) for each model. b) Ice surface profiles shown at 500-year intervals (Blue = older, Green = younger; Red line indicates the simulated present-day ice surface profile), with the underlying bed topography (black fill). The red tick mark on the x-axis denotes the present-day ice margin (Howat et al., 2014). Readers should refer to this figure for locations of flowlines used in this study.



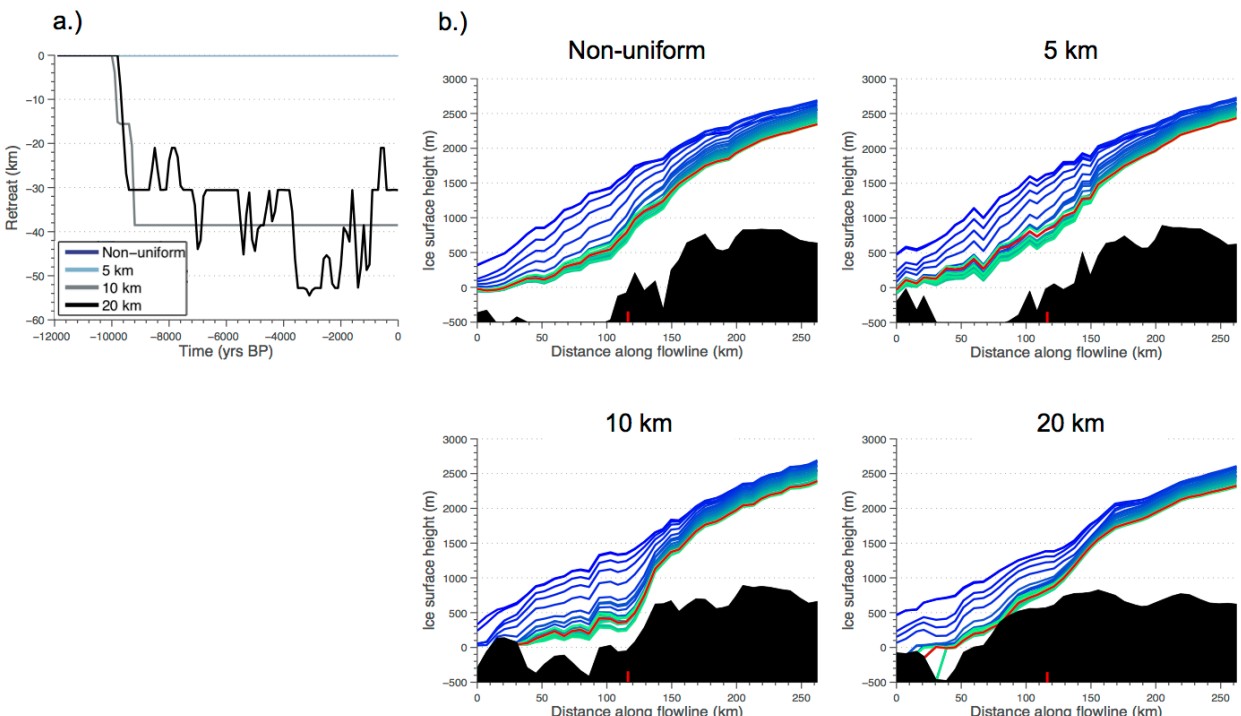

**Figure 9.** Same as in **Figure 8** but for flowline (D).