# Peer review of "The impact of model resolution on the simulated Holocene retreat of the Southwestern Greenland Ice Sheet using the Ice Sheet System Model (ISSM)"

_The Cryosphere, 2018_

## Referee Comment (RC1) · Robinson (Referee) · 5 Jan 2019

This study examines the effect of model resolution on simulating the Holocene retreat of southwest Greenland using ISSM. The study is well designed and does a good job of highlighting the impact of resolution on the results. The authors conclude that high resolution is particularly valuable in regions with complex bedrock terrain and fjords. I recommend publication after only minor revisions.

It is quite interesting how different retreat histories are obtained in the south with the

different resolution models. But then it also surprised me that none of the models actually compared well with geological constraints. Given this, it would be valuable to be able to visually compare the simulations to the constraints, for example, by adding some exposure ages of known locations to Figure 7.

I would suggest simplifying the Eqs. 2 and 4. Basal friction is important and the broader description here is relevant. However, it would be simpler to remove the exponent r and the term "|Vb|ˆs-1" since they are effectively not used. Also, the text for Eq. 4 is a bit ambiguous – does the limit of 300 apply to lambda or k? If it is applied to lambda, as it appears in Eq. 4, perhaps it would make more sense to apply this limit in Eq. 3 directly.

Finally, while the paper is generally well written, I would recommend additional proof-reading before resubmission. Some of the discussion seems repetitive, for example.

Minor comments

P3, line 18: of model resolution on => on model resolution for

P4, line 8-13: Consider rephrasing here (delete although).

P4, line 24-27: "Because . . ." <= consider removing this sentence or moving it to introduction, as this was already made clear and seems more related to the motivation.

P6, line 3: "Surface air temperatures . . . transiently" => "Transient surface air temperatures"

Reference: Tarbone et al. => Tabone et al.
* * *

---

## Referee Comment (RC2) · Seguinot (Referee) · 18 Jan 2019

As more computing time becomes available for ice sheet modelling, applications to long-term ice dynamics can afford higher and higher horizontal resolutions, so that an important question will need to be solved: how high is high enough? J. K. Cuzzone et al. present an application of the higher-order ice sheet model ISSM to the late deglaciation dynamics of the southwestern Greenland Ice Sheet margin. The authors present the results of a sensitivity test to model horizontal resolution. The model shows a different response to increased horizontal resolution in areas of different topographic

complexity, therefore bringing elements of answer to the aforementioned question.

The author's choice of model and study area make sense. While still an approximation of "full-Stokes" ice flow physics, the higer-order physics embedded in ISSM allows to escape some fundamental limitations associated with high-resolution applications of the Shallow ice approximation. Besides, applying ISSM to a paleo-glaciological context provides not only long-term (geological) validation data but, even more importantly, topographic data of much higher quality than is available under current ice sheets. The manuscript is logically organized and very well written.

I strongly support publication of these results, but provide below a list of comments (the first perhaps a bit far-fetched, the others mostly very specific) which the authors may use to complete the presentation of their model set-up and the discussion of their results.

**1 General comments**

**Which resolution is high enough?**

Although the high-resolution runs yield increased performance in regions of complex fjord topography, discrepancies with geological data remain. A potential explanation for these discrepancies is the lack of marine ice sheet processes such as calving and submarine melt in the model. However, they could also imply that even the high resolution of (up to) 2 km is not yet high enough. Do the model results allow to decipher between these two potential sources of error? Could you comment on this in your conclusions?

**Visualization of geological data**

The southwestern Greenland Ice Sheet margin is introduced as an area where the deglaciation history is well documented by the geological record. While differences

between the model results and this record are discussed, no visualization of the geological data is presented in the manuscript. If geological data is available, I think the manuscript would gain much by adding some of it in the figures.

**Repetitions in the discussion of model results**

I feel that parts of the results and discussion sections are repetitive, and that the manuscript could gain in clarity by minimizing these repetitions. Perhaps additional subsections in the discussion would help.

**2  Specific comments**

**p. 1, l. 22**: and one non-uniform

I suggest to replace with "and one using a non-uniform".

**p. 1, l. 28**: simulate unrealistic retreat

You could better highlight the main result in the abstract. Is the retreat unrealistically fast or too slow?

**p. 1, l. 32**: the SMB drives retreat

I suggest "the SMB predominantly drives retreat".

**p. 3, l. 11–12**: not well tested, however, is the sensitivity of simulated ice retreat to the ice flow dynamics model [...] and to model resolution
Such a sensitivity was conducted by Zekollari et al. (2017, Fig. 7). Comparative studies on multi-millenial time scales have also been published by Bernales et al. (2017); van Dongen et al. (2018).

**p. 4, l. 2–3**: model resolution is a constraint that is typically not explored when studying the past

I don't think this is correct. Although the results do not always appear in publication, I assume most modellers explore sensitivity to horizontal resolution. Some results are displayed for instance in Golledge et al. (2012, Fig. 6), Seguinot et al. (2016, Fig. 5), and Zekollari et al. (2017, Fig. 7).

**p. 4, l. 8**: We use the Ice Sheet System Model (ISSM)

For reproducibility, could you add a version number here?

**p. 4, l. 10**: this ice flow approximation is typically not used

Again you could refer to Zekollari et al. (2017) here.

**p. 4, l. 28–30**: coarse-resolution models do a poor job of capturing the complexities of the underlying topography (Aschwanden et al., 2016).

Although not exactly a paleo-ice sheet modelling study, I think it would be fair to mention here the grid sequencing approach used in spin-up by Aschwanden et al. (2016).

**p. 5, l. 2**: areas of smooth bed topography (primarily over the interior of the domain)

What do you mean with interior? Is this the area presently covered by ice and could the smooth bed topography be an artifact of the lack of data?
**p. 5, l. 5**: the GMIP DEM

I think this is "the GIMP DEM" (Greenland Ice Mapping Project).

**p. 5, l. 6**: In Figure 2

The order of figures 1 and 2 is inconsistent with the text.

**p. 5, l. 10**: Nuuk and Jakobshavn

These place names appear several times in the text. Could you please add them to at least one of the maps?

**p. 5, l. 15–16**: We use the positive degree day method outlined in Tarasov and Peltier (1999) to construct the necessary accumulation and ablation history

Could you please detail more precisely how your approach relates to theirs? The positive degree day method is used to compute ablation. The PDD method described in Tarasov and Peltier (1999) has been established in older references (see the first paragraph of Seguinot, 2013 for a concise review).

On the other hand, the accumulation scheme described by Tarasov and Peltier (1999) is more elaborate than most, but it seems not exactly consistent with the description of your accumulation model further in this paragraph. Am I correct?

**p. 5, l. 25**: a purely thermodynamic relationship as precipitation rate changes 7.3% for every 1°C

This formulation is unprecise, because it implies a different exponential factor depending if you consider a negative or positive change (92.7% is not the inverse of 107.3%). I would suggest an equation here or at least a reference. What do you mean with "a purely thermodynamic relationship"?

**p. 5, l. 27**: with allocation for superimposed ice

Do you allow for refreezing of melted snow/ice? If so, how is this parametrized, and is the surface mass balance model ran on yearly or sub-yearly time intervals?

**p. 5, l. 29–30**: elevation-dependent desertification is included

Presumably the temperature lapse-rate combined with temperature-dependent pre­cipitation reductions implies elevation-dependent precipitation reductions. Is this new mechanism applied on top or independently?

**p. 6, l. 15**: the spatially varying basal drag coefficient ($k$) in equation 2 is derived using inverse methods

Is this inversion performed independently for each horizontal resolution? From which resolution and how is the basal drag map otherwise interpolated?

**p. 6, l. 25–26**: a spatially varying temperature dependent scaling parameter ($\lambda$) as a function of time.

Could you please comment on the physical basis for this relation? Basal sliding is typically related to subglacial water pressure, while cold-based ice is theoretically as­sumed non-sliding irrelevant of its temperature. Does warming of cold-based ice below pressure melting induce unrealistic basal weakening in the model? Could this be the reason why a cap value is needed for $\lambda$, and is the scaling really an improvement over a constant friction map?

In the discussion section, could you add a few sentences to describe how the temper­ature evolves through the deglaciation simulations and how this affects the results?

**p. 7, l. 26**: For the regional model, we initialize the model with present day geometry

Could you please explain why the regional model is initialized in steady-state while transient boundary conditions from the global model are available? I wonder if this imply an inconsistence between the regional and global model initial states, where the global model's ice "remembers" the Last Glacial Maximum (temperature) whereas the regional model's ice do not.

**p. 7, l. 29**: the ice margin over southwestern Greenland was near or at the present-day coastline

Could you please detail what makes this condition an interesting starting point?

**p. 8, l. 1–2**: While grounding line migration is simulated in these experiments, calving and submarine melting of floating ice is not included.

How does floating ice then leaves the model? Is surface melting enough to constrain the ice shelves to somewhat reasonable size?

**p. 8, l. 26**: increasing ice volume with decreasing model resolution

Shouldn't this be "decreasing ice volume"? Low resolution yields low ice volume.

**p. 9, l. 21**: details in the ice margin similar in scale and sinuosity to the mapped ice moraines

A visualization of the mapped moraines would be very useful here.

**p. 12, l. 12–14**: small-scale marginal fluctuations are likely responsible for the moraine record and seem to be related to high-frequency variability in temperature

Are the fluctuations of the ice margin / volume in the model in phase with input temperature? Perhaps this could be visualized by plotting the input temperature on Fig. 6?

**p. 12, l. 24**: None of our experiments match the geologic observations.

Is it possible to say from your results, if this mismatch is due to physical processes (e.g. calving, submarine melt) missing from the model, or (still) too coarse resolution?

**p. 12, l. 29–31**: the ability of the high-resolution models to capture [...] is well captured.

There is one "capture" too many in this sentence.

**p. 14, l. 8**: Lecavalier et al. 2014)

The opening bracket is missing.

**Fig. 1**:

The caption is not visible in the manuscript.

**Fig. 2**:

For those non familiar with unstructured meshes (like me), would it be possible to display the mesh used in each experiment (esp. experiment A), or is it too dense to show? For visualization puposes I would also recommend to use the same inline arrangement of figure panels as in Fig. 3 and 6.

**Fig. 3**:

To save space I think you could remove the observed velocities (already shown on Fig. 1, and too similar to those modelled using the non-uniform mesh for a direct comparison).

**Fig. 6**:

I think there is a problem with the labelling of the x-axis. The unit "yrs BP" corresponds to an age which should be a positive number. Either remove the minus signs and replace "time" by "age", or remove the BP (similary on Fig. 8 and 9).

Is it possible to display the input temperature forcing alongside the model output? I think this would help one understand the short-term fluctuations in ice volume as well as the longer-term deglaciation dynamics.

Besides, this might be a personal preference but I find the min-max normalization not so informative. I understand that the different runs start with different ice volumes because of their resolutions, but I find it difficult to know what I am looking at after normalization. I would personally omit panel B and instead combine panel A with Fig. 4. Then, one quickly understands the different initial ice volumes.

**Fig. 7**:

I do not understand the choice of separate panels A and B with (only a small) area missing in between. It would be more visual to combine them. If space is to be saved, more ocean and modern ice area can be cropped.

Is geological data on the mapped moraine available to be plotted alongside the model results?

Congratulations again on your work and I hope you will find my comments useful in bringing your manuscript to final form.

**References**

Aschwanden, A., Fahnestock, M. A., and Truffer, M.: Complex Greenland outlet glacier flow captured, Nature Communications, 7, doi:10.1038/ncomms10524, 2016.

Bernales, J., Rogozhina, I., Greve, R., and Thomas, M.: Comparison of hybrid schemes for the combination of shallow approximations in numerical simulations of the Antarctic Ice Sheet, The Cryosphere, 11, 247–265, doi:10.5194/tc-11-247-2017, 2017.

Golledge, N. R., Mackintosh, A. N., Anderson, B. M., Buckley, K. M., Doughty, A. M., Barrell, D. J., Denton, G. H., Vandergoes, M. J., Andersen, B. G., and Schaefer, J. M.: Last Glacial Maximum climate in New Zealand inferred from a modelled Southern Alps icefield, Quaternary Res., 46, 30–45, doi:10.1016/j.quascirev.2012.05.004, 2012.

Seguinot, J.: Spatial and seasonal effects of temperature variability in a positive degree-day glacier surface mass-balance model, J. Glaciol., 59, 1202–1204, doi:10.3189/2013JoG13J081, 2013.

Seguinot, J., Rogozhina, I., Stroeven, A. P., Margold, M., and Kleman, J.: Numerical simulations of the Cordilleran ice sheet through the last glacial cycle, The Cryosphere, 10, 639–664, doi:10.5194/tc-10-639-2016, 2016.

Tarasov, L. and Peltier, W. R.: Impact of thermomechanical ice sheet coupling on a model of the 100 kyr ice age cycle, Journal of Geophysical Research: Atmospheres, 104, 9517–9545, doi:10.1029/1998jd200120, 1999.

van Dongen, E. C. H., Kirchner, N., van Gijzen, M. B., van de Wal, R. S. W., Zwinger, T., Cheng, G., Lötstedt, P., and von Sydow, L.: Dynamically coupling full Stokes and shallow shelf approximation for marine ice sheet flow using Elmer/Ice (v8.3), Geoscientific Model Development, 11, 4563–4576, doi:10.5194/gmd-11-4563-2018, 2018.

Zekollari, H., Lecavalier, B. S., and Huybrechts, P.: Holocene evolution of Hans Tausen Iskappe (Greenland) and implications for the palaeoclimatic evolution of the high Arctic, Quaternary Science Reviews, 168, 182–193, doi:10.1016/j.quascirev.2017.05.010, 2017.
* * *

---

## Author Comment (AC1) · 18 Feb 2019

**REVIEWER #1, Alexander Robinson**

- This study examines the effect of model resolution on simulating the Holocene retreat of southwest Greenland using ISSM. The study is well designed and does a good job of highlighting the impact of resolution on the results. The authors conclude that high resolution is particularly valuable in regions with complex bedrock terrain and fjords. I recommend publication after only minor revisions.

We would like to thank Alexander Robinson for his constructive review and input.

- It is quite interesting how different retreat histories are obtained in the south with the different resolution models. But then it also surprised me that none of the models actually compared well with geological constraints. Given this, it would be valuable to be able to visually compare the simulations to the constraints, for example, by adding some exposure ages of known locations to Figure 7.

Regarding the first point. We do not necessarily find it surprising that the models do not capture the retreat history well. The models are driven with the commonly used oxygen isotope scaling, which scales present day precip. and temperature along the NGRIP oxygen isotope curve. Therefore, the climatology used to force the model through time: 1.) Uses info from a single location (NGRIP) which is likely colder than the actual temperature anomalies experienced in more distal locations in our regional model and 2.) This anomaly is applied as spatially constant forcing, which is also a crude approximation that does not account for spatial variations in temperature.

We also want to clarify that the goal of this work was not to match the geologic retreat history. This is actually the goal of ongoing work which is part of a larger project to refine and add additional geologic constraints on past retreat. Future simulations will include improved climate forcings (temp. and precip.) and will investigate the role basal melting on floating ice may play in the simulated behavior of the ice sheet in the fjord regions.

We have modified Figure 6 by adding a summary of our current understanding of the ice retreat history in figure 6 which we hope will aid the readers (From Lesnek and Briner, 2018).

- I would suggest simplifying the Eqs. 2 and 4. Basal friction is important and the broader description here is relevant. However, it would be simpler to remove the exponent r and the term "|Vb|^s-1" since they are effectively not used. Also, the text for Eq. 4 is a bit ambiguous – does the limit of 300 apply to lambda or k? If it is applied to lambda, as it appears in Eq. 4, perhaps it would make more sense to apply this limit in Eq. 3 directly. Finally, while the paper is generally well written, I would recommend additional proofreading before resubmission. Some of the discussion seems repetitive, for example.

We have made changes to equations 2 and 4 following the reviewer's suggestions.

We noticed a mistake in our equation 4. We have changed it from:

$$\tau_b = -\min(300, \lambda_t)\, k^2 N^r \left|\left|V_b\right|\right|^{s-1} v_b$$

To:

$$\tau_b = -\lambda_t \min(300, k^2) \, N^r \left\| V_b \right\|^{s-1} v_b$$

Here, we are capping k, the friction coefficient at 300. Basically, any value above 250 corresponds to no sliding, so we cap this at 300 to avoid numerical instabilities that may arise with very high values of k.

As recommended by the reviewer #2, we have moved certain portions of the discussion into their own sub-sections and have removed repetitive statements.

Minor comments
- P3, line 18: of model resolution on => on model resolution for

Done

- P4, line 8-13: Consider rephrasing here.

We have adjusted this to read:

"Although recent work has used the higher order approximation in simulations over past time periods (Zekollari et al., 2017), this ice flow approximation still remains not commonly used when simulating over paleoclimate timescales. We use this approximation, however, as our choice is based upon representing the past dynamics of the ice sheet history as best as possible even though computational time is increased over conventional paleoclimate ice sheet models using the more common shallow ice approximation (SIA; Hutter, 1983). "

- P4, line 24-27: "Because . . ." <= consider removing this sentence or moving it to introduction, as this was already made clear and seems more related to the motivation.

We have decided to remove this sentence.

- P6, line 3: "Surface air temperatures . . . transiently" => "Transient surface air temperatures"

Done

- Reference: Tarbone et al. => Tabone et al.

Done

**REVIEWER #2, Julien Seguinot**

As more computing time becomes available for ice sheet modelling, applications to long-term ice dynamics can afford higher and higher horizontal resolutions, so that an important question will need to be solved: how high is high enough? J. K. Cuzzone et al. present an application of the higher-order ice sheet model ISSM to the late deglaciation dynamics of the southwestern Greenland Ice Sheet margin. The authors present the results of a sensitivity test to model horizontal resolution. The model shows a different response to increased horizontal resolution in areas of different topographic C1 complexity, therefore bringing elements of answer to the aforementioned question.

The author's choice of model and study area make sense. While still an approximation of "full-Stokes" ice flow physics, the higher-order physics embedded in ISSM allows to escape some fundamental limitations associated with high-resolution applications of the Shallow ice

approximation. Besides, applying ISSM to a paleo-glaciological context provides not only long-term (geological) validation data but, even more importantly, topographic data of much higher quality than is available under current ice sheets. The manuscript is logically organized and very well written.

I strongly support publication of these results, but provide below a list of comments (the first perhaps a bit far-fetched, the others mostly very specific) which the authors may use to complete the presentation of their model set-up and the discussion of their results.

We would like to thank Julien Seguinot for his constructive input and review.

**General comments**
Which resolution is high enough? Although the high-resolution runs yield increased performance in regions of complex fjord topography, discrepancies with geological data remain. A potential explanation for these discrepancies is the lack of marine ice sheet processes such as calving and submarine melt in the model. However, they could also imply that even the high resolution of (up to) 2 km is not yet high enough. Do the model results allow to decipher between these two potential sources of error? Could you comment on this in your conclusions?

We agree and also discuss in "Discussion section 4.2: Model Limitations" that mismatch between our high-resolution model and the fjord retreat captured in the geologic record may be due not only to the surface forcing alone, but may include a component of basal melting of floating ice and calving.

We are currently working on simulations for another project that investigate the role of basal melting of floating ice, to determine if we can reach better agreement with the geologic record  As stated in the response to Reviewer #1, the point of this work was to not match the geologic record. We offer no tuning or sensitivity runs to try and achieve a match with the geologic reconstruction. This is the focus of ongoing and future work.

We also agree that 2 km resolution may be insufficient to capture the grounding line migration accurately and research suggests (Seroussi and Morlighem, 2018) that 1 km resolution or lower is necessary to accurately capture grounding line changes.  An issue exists however, going sub-1 km resolution in that information regarding the bedrock geometry rarely goes down to such high resolution. So even with higher resolution meshes in fjord locations, a lack of information regarding the bedrock geometry in fjord regions will also lead to uncertainties in simulated changes in the grounding line and therefore the ice margin.  For these simulations, 2 km is the lowest resolution that can currently be performed over such long timescales.

Ultimately, however, our conclusions support that fjord geometry is not well captured in low resolution models >10km.  The retreat simulated in the low-resolution models in fjord regions only occurs as the fjord geometry and primarily depth of the fjord is not resolved.

We have added a statement in the 'Discussion section 4.2 Model Limitations:

"Additionally, in applying basal melting to floating ice, it is uncertain whether 2 km resolution would be sufficient to accurately capture grounding line migration as recent research (Seroussi and Morlighem, 2018) suggests that resolutions of 1 km or higher are often necessary to match present day fluctuations."

**Visualization of geological data**
The southwestern Greenland Ice Sheet margin is introduced as an area where the deglaciation history is well documented by the geological record. While differences between the model results and this record are discussed, no visualization of the geological data is presented in the manuscript. If geological data is available, I think the manuscript would gain much by adding some of it in the figures.

The research presented in this paper is part of a larger, current project, which is working to improve the chronology of retreat across Southwestern Greenland.  This work entails a large number (>80) of cosmogenic exposure ages using Be[10] that refine and map the retreat history in fine detail, both spatially and temporally.   Nevertheless, we agree that context of the disagreement would be helpful for the reader in map form.   We have updated figure 6 to include a summary of the current understanding of the retreat history across our domain from Lesnek and Briner (2018).

More importantly, the goal of the experiments presented here were not to achieve a match with the current geologic chronology of past ice retreat.  To do so would mean much more consideration and attempts to model sensitivities to past climate, which is the focus of our current modeling efforts.  Our goal with this paper was to outline the differences in ice retreat across a model with varying resolutions.

**Repetitions in the discussion of model results**
I feel that parts of the results and discussion sections are repetitive, and that the manuscript could gain in clarity by minimizing these repetitions. Perhaps additional subsections in the discussion would help.

As recommended by the reviewer, we have moved certain portions of the discussion into their own sub-sections and have removed repetitive statements.

- p. 1, l. 22: and one non-uniform
I suggest to replace with "and one using a non-uniform".

Done

- p. 1, l. 28: simulate unrealistic retreat
You could better highlight the main result in the abstract. Is the retreat unrealistically fast or too slow?

We have adjusted the sentence pg.1, line 28, removing unrealistic.
……simulate "retreat that occurs as" ice-surface lowering….

The retreat is unrealistic in that it is driven by a failure of the coarse mesh to capture deep fjords. Therefore, the retreat in this case is too fast.  It is expected that by increasing the temperature forcing, surface lowering would occur at a faster pace, and therefore retreat would occur faster as well.

- p. 1, l. 32: the SMB drives retreat
I suggest "the SMB predominantly drives retreat".

Done

- p. 3, l. 11–12: "not well tested, however, is the sensitivity of simulated ice retreat to the ice flow dynamics model [...] and to model resolution"
Such a sensitivity was conducted by Zekollari et al. (2017, Fig. 7). Comparative studies on multi-millenial time scales have also been published by Bernales et al. (2017); van Dongen et al. (2018).

Thank you for the paper recommendations.  The Zekollari et al. (2017) paper is a great citation for dealing with model formulation and resolution over paleoclimate time periods.  We have adjusted the text:

From:
An area within paleoclimate ice sheet modeling that remains not well tested, however, is the sensitivity of simulated ice retreat to the ice flow dynamics model (i.e. the level of complexity in its numerical approximations) and to model resolution, both in time and space.

To
Although recent experiments have investigated sensitivities to model formulation (Zekollari et  al., 2017) and horizontal resolution over past climates (Zekollari et  al., 2017; Seguinot et al., 2016; Golledge et al., 2012), testing the sensitivity of simulated ice retreat to the ice flow dynamics model (i.e. the level of complexity in its numerical approximations) and to model resolution, both in time and space still remains an important area of research.

- p. 4, l. 2–3: "model resolution is a constraint that is typically not explored when studying the past"
I don't think this is correct. Although the results do not always appear in publication, I assume most modellers explore sensitivity to horizontal resolution. Some results are displayed for instance in Golledge et al. (2012, Fig. 6), Seguinot et al. (2016, Fig. 5), and Zekollari et al. (2017, Fig. 7).

On page 3, l. 11-12 we added citations to these papers expressing that they have examined the impact of mesh resolution.   The statement here highlights that it is still an area that requires more research when applied to studying past climates.

- p. 4, l. 8: "We use the Ice Sheet System Model (ISSM)"
For reproducibility, could you add a version number here?

Added V4.13

- p. 4, l. 10: "this ice flow approximation is typically not used"
Again you could refer to Zekollari et al. (2017) here.

Thank you for the recommendation.

We have changed these sentences to read:

"Although recent work has used the higher order approximation in simulations over past time periods (Zekollari et al., 2017), this ice flow approximation still remains not commonly used when simulating over paleoclimate timescales. We use this approximation, however, as our choice is based upon representing the past dynamics of the ice sheet history as best as possible even though computational time is increased over conventional paleoclimate ice sheet models using the more common shallow ice approximation (SIA; Hutter, 1983). "

- p. 4, l. 28–30: coarse-resolution models do a poor job of capturing the complexities of the underlying topography (Aschwanden et al., 2016).
Although not exactly a paleo-ice sheet modelling study, I think it would be fair to mention here the grid sequencing approach used in spin-up by Aschwanden et al. (2016).

Upon request from Reviewer #1 we removed this sentence since we already introduced the role grid resolution plays in correctly capturing contemporary ice velocities in the introduction.

- p. 5, l. 2: areas of smooth bed topography (primarily over the interior of the domain)
What do you mean with interior? Is this the area presently covered by ice and could the smooth bed topography be an artifact of the lack of data?

We agree that there is a lack of data which limits our current understanding of the sub ice bed topography in the interior of the ice sheet. We corrected our language in the sentence to reflect that our grid resolution becomes more coarse in areas where gradients in the bed topography are more smooth. This is consistent with the information that we have from mass conservation methods (BedMachine; Morlighem et al, 2017).

We have changed the sentence from:

"The maximum horizontal mesh resolution is 15 km in areas of smooth bed topography (primarily over the interior of the domain) and becomes progressively finer in areas of high relief, with a minimum horizontal resolution of 2 km (mainly in fjord regions)."

To:

"The maximum horizontal mesh resolution is 15 km where gradients in the bed topography are smooth (primarily over the interior of the domain) and becomes progressively finer in areas of high relief, with a minimum horizontal resolution of 2 km (mainly in fjord regions).

- p. 5, l. 5: "the GMIP DEM"
I think this is "the GIMP DEM" (Greenland Ice Mapping Project).
Thanks for the correction. Done.

- p. 5, l. 6: In Figure 2
The order of figures 1 and 2 is inconsistent with the text.

It should already be consistent with the text. Figure 1 is introduced earlier on page 4, around line14.

- p. 5, l. 10: "Nuuk and Jakobshavn"
These place names appear several times in the text. Could you please add them to at least one of the maps?

These locations were added to Figure 1.  Unfortunately, in the PDF uploaded for discussions, the figure blocked the caption.  We apologize about this, but the latest version will include this fix.

- p. 5, l. 15–16: We use the positive degree day method outlined in Tarasov and Peltier(1999) to construct the necessary accumulation and ablation history

Could you please detail more precisely how your approach relates to theirs? The positive degree day method is used to compute ablation. The PDD method described in Tarasov and Peltier (1999) has been established in older references (see the first paragraph of Seguinot, 2013 for a concise review). On the other hand, the accumulation scheme described by Tarasov and Peltier (1999) is more elaborate than most, but it seems not exactly consistent with the description of your accumulation model further in this paragraph. Am I correct?

Our approach follows directly from Tarasov and Peltier (1999).  The accumulation scheme follows equation 5 in Tarasov and Peltier, 1999.  The inputs are a temperature and precipitation record, which we use the GRIP oxygen isotope record to scale through time.  We reference Le Morzadec et al., 2015 which has a more extensive review of the scheme and its implementation in ISSM (see Le Morzadec et al., 2015 supplement).

- p. 5, l. 25: "a purely thermodynamic relationship as precipitation rate changes 7.3% for every 1_C"

This formulation is unprecise, because it implies a different exponential factor depending if you consider a negative or positive change (92.7% is not the inverse of 107.3%).
I would suggest an equation here or at least a reference. What do you mean with "a purely thermodynamic relationship"?

By "a purely thermodynamic relationship," we mean it follows the Clausius-Clapeyron relationship. There is no additional information that we used to constrain dynamic changes (atmospheric circulation) during the past. The 7.3% for every 1 degree is taken from Huybrechts, 2002.

We have changed the sentence:
"Deviations in the precipitation climatology arise from a purely thermodynamic relationship as precipitation rate changes 7.3% for every 1°C of temperature change derived in equation 1."

To :
"Precipitation rate changes 7.3% for every 1°C of temperature change derived in equation 1 (Huybrechts, 2002)."

- p. 5, l. 27: "with allocation for superimposed ice"
Do you allow for refreezing of melted snow/ice? If so, how is this parametrized, and is the surface mass balance model ran on yearly or sub-yearly time intervals?

We allow for refreezing following Janssens and Huybrechts (2000). We have added a pointer to "(see supplemental information in Le Morzadec et al., 2015) which outlines this process. The smb model is resolved on monthly timesteps.

- p. 5, l. 29–30: "elevation-dependent desertification is included"

Presumably the temperature lapse-rate combined with temperature-dependent precipitation reductions implies elevation-dependent precipitation reductions. Is this new mechanism applied on top or independently?

The lapse rate adjustment will have the effect of lowering the precipitation as a function of the Clausius-Clapeyron relationship (thermodynamic). The elevation-dependent precip. reduction tries to capture "rain shadowing" effect of increasing ice topography. The desertification effect is applied independent on the lapse rate corrected precip.

- p. 6, l. 15: "the spatially varying basal drag coefficient (k) in equation 2 is derived using inverse methods"

Is this inversion performed independently for each horizontal resolution? From which resolution and how is the basal drag map otherwise interpolated?

Yes, the inversion is performed independently for each horizontal resolution.

- p. 6, l. 25–26: "a spatially varying temperature dependent scaling parameter ($\lambda$) as a function of time."

Could you please comment on the physical basis for this relation? Basal sliding is typically related to subglacial water pressure, while cold-based ice is theoretically assumed non-sliding irrelevant of its temperature. Does warming of cold-based ice below pressure melting induce unrealistic basal weakening in the model? Could this be the reason why a cap value is needed for, and is the scaling really an improvement over a constant friction map? In the discussion section, could you add a few sentences to describe how the temperature evolves through the deglaciation simulations and how this affects the results?

The temperature dependent sliding has been implemented in Paleo ice-sheet modeling experiments for some time now, and we follow the approach of Hindmarsh and LeMeur (2001) and Greve (2005). In this parameterization, we only scale the friction coefficient. When temperatures decrease, the friction will increase following the relationship outlined in the methods section (visa versa for warming). A friction coefficient ~250 implies cold based ice, or virtually no sliding. ISSM inverts for the present-day friction coefficient that provides the best match to present day ice surface velocities. In this case, the friction coefficient is constrained for modern day, and therefore the sliding regime too. We assume that during the much colder Younger Dryas (12 ka), upon which we relax our ice model, subglacial melt was lower than present day – which should correspond to less sliding than present day – so in ISSM this would require a higher friction coefficient. In our model relaxation at 12 ka, basal temperatures, particularly in outlets are lower than present day. Using the friction scaling scheme, lower basal temperatures equate to a higher friction coefficient, less sliding, and therefore ice sheet growth during our relaxation. As the climate warms through time, basal temperatures in thinner regions of the ice sheet (ice streams and marginal ice) warm and sliding increases.

Warming below the pressure melting point does not create unrealistic basal weakening in our model. Instead, internal deformation of the ice increases and the sliding increases due to the scheme implemented. However, when the ice is not close to the pressure melting point, the sliding is limited. The cap value is only implemented as virtually no sliding occurs with friction coefficient values above 250. Therefore, we prevent any numerical instabilities that could occur from values much higher than this. This scheme has been outlined and implemented within a number of paleo ice-sheet models cited above.

We have added a statement in Section 4.4 (Model Limitations) pg. 14, line 21 addressing this:

"Additionally, the scaling of the basal friction coefficient introduces some uncertainties particularly when considering the temperature forcing throughout the simulation. Our method for scaling the basal friction coefficient through time follows a common approach used in many modeling studies over paleoclimate timescales (Hindmarsh and LeMeur, 2001; Greve ,2005). For these simulations, the evolution of the basal temperatures through time depends on the surface temperature forcing which is derived from the GRIP $\delta^{18}O$ scaling. Therefore, changes to the surface temperature forcing can impact the evolution of the basal temperatures over time, which ultimately affects the ice sliding following this approach. This model limitation falls within the bounds of current ice sheet modeling efforts whereby a lack of physically based basal sliding parameterizations exist. Despite this limitation, the conclusions presented here remain unaffected."

- p. 7, l. 26: "For the regional model, we initialize the model with present day geometry"

Could you please explain why the regional model is initialized in steady-state while transient boundary conditions from the global model are available? I wonder if this imply an inconsistence between the regional and global model initial states, where the global model's ice "remembers" the Last Glacial Maximum (temperature) whereas the regional model's ice do not.

We relax the regional model using a constant 12 ka climate, and boundary conditions of temperature, ice velocity, and thickness from the global model. In this case we do not interpolate the global model onto the regional mesh, but instead "grow" the ice sheet from a present-day geometry out to the domain edge using the boundary conditions from the global model at 12 ka.

The reviewer makes a good point that the memory from the LGM may not be represented in the regional model. This however does not affect our results and conclusions, but is a consideration that we will look into for our present and future work.

- p. 7, l. 29: "the ice margin over southwestern Greenland was near or at the present-day coastline"

Could you please detail what makes this condition an interesting starting point?

We have added an additional statement to this sentence:

"This time period is chosen as the ice margin over southwestern Greenland was near or at the present-day coastline with the margin remaining stable during this interval (Young and Briner, 2015)."

- p. 8, l. 1–2: While grounding line migration is simulated in these experiments, calving and submarine melting of floating ice is not included. How does floating ice then leaves the model? Is surface melting enough to constrain the ice shelves to somewhat reasonable size?

Where the ice margin is located at the domain edge, we have a free-flux condition, where mass exiting the domain is lost. In this case floating ice will only respond to surface driven melt if the ice stream does not make it to the domain edge. This is likely a reason why ice persists in fjord regions for the higher resolution models. It would likely require calving or submarine melting to melt the floating ice completely.

- p. 8, l. 26: "increasing ice volume with decreasing model resolution"
Shouldn't this be "decreasing ice volume"? Low resolution yields low ice volume.

Yes, you are right. We have changed this.

- p. 9, l. 21: "details in the ice margin similar in scale and sinuosity to the mapped ice moraines"

A visualization of the mapped moraines would be very useful here.

We have updated figure 6 to show a summary of the geologic retreat history and the respective chronology from Lesnek and Briner (2018).

This figure in the paper cited nicely illustrates the mapped moraine sequences which is under scrutiny with the chronology currently being refined and developed.

- p. 12, l. 24: "None of our experiments match the geologic observations."

Is it possible to say from your results, if this mismatch is due to physical processes (e.g. calving, submarine melt) missing from the model, or (still) too coarse resolution?

This is the focus of current work on project with glacial geologists who are adding additional constraints and refining the current chronology of retreat. On important constraint is that we are forcing our climatology using the NGRIP isotope record. Therefore, this forcing is likely too cold as it assumes all variations in temperature are controlled by one record (NGRIP) and does not account for past spatial variation in temperature (anomaly applied spatially constant). Future runs will use improved temperature and precipitation reconstructions as forcings. Future runs will also apply submarine melting on floating ice. Therefore, I assume mismatch is certainly influenced not only by a lack of calving or submarine melt in these simulations, but from a poor climate forcing. From our results the retreat was similar in the Northern portion of our domain regardless of SMB. We highlighted that in these areas, resolution would be secondary to getting the SMB right.

- p. 12, l. 29–31: the ability of the high-resolution models to capture [...] is well captured.
There is one "capture" too many in this sentence.

We have changed the first "capture" to "resolve."

Fixed, thanks.

- Fig. 1:
The caption is not visible in the manuscript.

We will make sure this is clear in the resubmitted version.

- Fig. 2:
For those non familiar with unstructured meshes (like me), would it be possible to display the mesh used in each experiment (esp. experiment A), or is it too dense to show? For visualization pupuses I would also recommend to use the same inline arrangement of figure panels as in Fig. 3 and 6.

We have added a note (pg. 5, line 2) "see Figure S1 for visualization of mesh resolutions."  We have added a supplemental figure (S1) that shows the different meshes.

- Fig. 3:
To save space I think you could remove the observed velocities (already shown on Fig. 1, and too similar to those modelled using the non-uniform mesh for a direct comparison).

Thanks for the recommendation.  We have made this change.

- Fig. 6:
I think there is a problem with the labelling of the x-axis. The unit "yrs BP" corresponds to an age which should be a positive number. Either remove the minus signs and replace "time" by "age", or remove the BP (similary on Fig. 8 and 9). Is it possible to display the input temperature forcing alongside the model output? I think this would help one understand the short-term fluctuations in ice volume as well as the longer-term deglaciation dynamics. Besides, this might be a personal preference but I find the min-max normalization not so informative. I understand that the different runs start with different ice volumes because of their resolutions, but I find it difficult to know what I am looking at after normalization. I would personally omit panel B and instead combine panel A with Fig. 4. Then, one quickly understands the different initial ice volumes.

We have changed the x-axis following the recommendation.   Also, we now omit the Min-Max normalized curves.

- Fig. 7:
I do not understand the choice of separate panels A and B with (only a small) area missing in between. It would be more visual to combine them. If space is to be saved, more ocean and modern ice area can be cropped. Is geological data on the mapped moraine available to be plotted alongside the model results?

We chose to focus on these sub regions because we strongly think that it provides the reader more focus, and highlights the distinction between the retreat in the north and the retreat in the south. We have played around with showing the full domain, but by focusing, we feel this will guide the

reader much better. We have modified figure 6 to include the current understanding of the geologic retreat and the associated chronology which has come from Lesnek and Briner (2018).

Congratulations again on your work and I hope you will find my comments useful in bringing your manuscript to final form.

We greatly appreciate your encouraging remarks and detailed review. Thank you again for the thorough and constructive review.

**References:**

Greve, R., Saito, F., and Abe-Ouchi. A. Initial results of the SeaRISE numerical experiments with the models SICOPOLIS and IcIES for the Greenland Ice Sheet, Ann. Glaciol., 52, 23–30, https://doi.org/10.3189/172756411797252068, 2011.

Hindmarsh, R. C. A. and E. Le Meur. Dynamical processes involved in the retreat of marine ice sheets. J. Glaciol., 47 (157), 271–282, 2001.

Janssens, I. and Huybrechts, P. (2000): The treatment of meltwater retention in mass-balance parameterisations of the Greenland ice sheet , Annals of Glaciology, 31 , pp. 133-140 .

Lesnek, A, J., Briner, J. Response of a land-terminating sector of the western Greenland Ice Sheet to early Holocene climate change: Evidence from $^{10}$Be dating in the Søndre Isortoq region. Quat. Sci. Rev.  180, 145-156. https://doi.org/10.1016/j.quascirev.2017.11.028, 2018.

Seroussi, H. and Morlighem, M.: Representation of basal melting at the grounding line in ice flow models, The Cryosphere, 12, 3085-3096, https://doi.org/10.5194/tc-12-3085-2018, 2018.